# Fluid Overload and Tissue Sodium Accumulation as Main Drivers of Protein Energy Malnutrition in Dialysis Patients

**DOI:** 10.3390/nu14214489

**Published:** 2022-10-25

**Authors:** Bernard Canaud, Marion Morena-Carrere, Helene Leray-Moragues, Jean-Paul Cristol

**Affiliations:** 1School of Medicine, Montpellier University, 34000 Montpellier, France; 2Global Medical Office, FMC-France, 94260 Fresnes, France; 3PhyMedExp, Department of Biochemistry and Hormonology, INSERM, CNRS, University Hospital Center of Montpellier, University of Montpellier, 34000 Montpellier, France; 4Charles Mion Foundation, AIDER-Santé, 34000 Montpellier, France

**Keywords:** sodium, water, fluid overload, fluid depletion, dialysis adequacy, malnutrition

## Abstract

Protein energy malnutrition is recognized as a leading cause of morbidity and mortality in dialysis patients. Protein–energy-wasting process is observed in about 45% of the dialysis population using common biomarkers worldwide. Although several factors are implicated in protein energy wasting, inflammation and oxidative stress mechanisms play a central role in this pathogenic process. In this in-depth review, we analyzed the implication of sodium and water accumulation, as well as the role of fluid overload and fluid management, as major contributors to protein–energy-wasting process. Fluid overload and fluid depletion mimic a tide up and down phenomenon that contributes to inducing hypercatabolism and stimulates oxidation phosphorylation mechanisms at the cellular level in particular muscles. This endogenous metabolic water production may contribute to hyponatremia. In addition, salt tissue accumulation likely contributes to hypercatabolic state through locally inflammatory and immune-mediated mechanisms but also contributes to the perturbation of hormone receptors (i.e., insulin or growth hormone resistance). It is time to act more precisely on sodium and fluid imbalance to mitigate both nutritional and cardiovascular risks. Personalized management of sodium and fluid, using available tools including sodium management tool, has the potential to more adequately restore sodium and water homeostasis and to improve nutritional status and outcomes of dialysis patients.

## 1. Introduction

Malnutrition is a common disorder in dialysis patients [1,2,3]. Protein–energy malnutrition (PEM) or protein–energy-wasting (PEW) process is the most commonly observed manifestation of this disorder [4]. As stated, in PEW process, malnutrition affects the two main components of body composition, namely protein stores (muscle mass, sarcopenia) and energy stores (fat mass) with different behaviors and/or severities [5]. According to recent reviews, the prevalence of PEM in dialysis patients may range between 15 and 75% [6]. This imprecise estimate reflects several factors that are poorly considered, such as criteria used to define PEM, variability of existing assessment tools, age-related factors, geographical differences, stages of chronic kidney disease, and effects of renal replacement modalities [4,7,8,9]. Chronic kidney disease progression is associated with a worsening of protein energy malnutrition, culminating in end-stage kidney disease and in dialysis patients [10]. This deleterious phenomenon has been clearly highlighted within the MDRD study and directly linked to the aggravation of uremic disorders marked by a spontaneous reduction of diet caloric and protein intakes [11,12]. In a recent international systematic review by Carrero et al., it was shown that prevalence of PEM in dialysis patients averaged 45% worldwide but with large variations according to country incomes [6]. As clearly identified in this review, prevalence of PEM was 39%, 48%, and 63% in high-, middle-, and low-income countries, respectively [6].

Protein–energy malnutrition in dialysis patients is most often characterized by inflammation and oxidative stress mechanisms [13,14,15,16,17,18]. Inflammation was identified early by researchers of Karolinska’s group as a prominent and common marker of malnutrition and atherosclerosis process in dialysis patients, the so-called malnutrition inflammation atherosclerosis (MIA) syndrome [19,20]. Later on, subclinical chronic inflammation was confirmed by independent researchers as being associated in a broader pathophysiologic spectrum called malnutrition inflammation complex syndrome (MICS) [13,14].

Protein energy malnutrition is now a well-recognized leading cause of morbidity and mortality in dialysis patients [14,21]. In that perspective, nutritional consensus expert groups have drawn attention of the clinical community to this matter in order to monitor the nutritional status of dialysis patients more closely and to use a panel of appropriate tools, including inflammatory markers (CRP), in order to intervene in a timely appropriate fashion to prevent further deterioration and poor outcomes [4,22]. To characterize protein–energy malnutrition, a panel of three to four biomarkers is usually required. Without going into detail, biomarkers include hypoalbuminemia, reduced diet caloric and protein intakes (loss of appetite, diet survey), change in body composition (subjective global assessment, unintended weight loss, anthropometric, muscle bioimpedance, loss of muscle mass), and CRP increase [4]. Protein energy malnutrition is associated with higher hospitalization risk, including risk of infections, falls, fractures, vascular access dysfunction, and poor healing after surgery [23,24]. Furthermore, mortality from all causes, but particularly from cardiovascular and infection origins, is significantly increased with the degree of protein–energy malnutrition [25].

## 2. Protein Energy Malnutrition Results from Several Causes

Protein–energy malnutrition has multiple causes in dialysis patients that may act individually but most likely tend to act in combination either simultaneously or sequentially [26,27]. For practical reasons, causes of PEM are analyzed individually and in a more integrated vision. Schematically, they belong to four groups of factors: patient-related factors, uremic-related factors, dialysis-related factors, and intercurrent-illness-related-factors. They are summarized in Figure 1.

Patient-related factors depend mainly on age, gender, life and diet habits, socio-economic conditions, cultural assets, past medical history (long standing history of disease and treatment), and comorbid conditions (diabetes, cardiac or liver disease). Most of them are fixed factors out of medical reach [27,28].

Uremic-related factors consist of all disorders associated with kidney failure that impact negatively nutritional status. Uremia is a complex disorder involving multiple factors that act synergistically to impede nutritional status [29]. To facilitate such analysis, we propose categorizing them according to their two main pathophysiologic pathways: firstly, anti-anabolic factors; secondly, pro-catabolic factors. This is schematized in Figure 2. As presented, on one hand, antianabolic elements include factors that tend to reduce nutrient intake or gut absorption (anorexia, gastroparesis, satiety, taste, medication) [30,31], contributing to functional deficiencies and metabolic or endocrine disorders that impede cell or tissular processing of nutrients (vitamin deficiency, protein and amino acid deficiency, hormonal deficiency, receptor impingement, anemia) [32,33,34,35]. On the other hand, procatabolic factors include metabolic acidosis, fluid overload, inflammation, oxidative stress, hyperparathyroidism, and uremic toxin accumulation that have cell and mitochondrial toxicity [36,37].

Dialysis-related factors are superimposed conditions to uremia with dual and opposite action. On one hand, hemodialysis has a positive action by controlling uremic disorders and restoring internal milieu homeostasis through periodic reduction of nitrogenous waste products and correction of electrolytic and fluid volume imbalance. On the other hand, hemodialysis has a negative action by creating unphysiological cyclic shifts and by triggering directly protein muscle degradation to compensate for amino acid, peptides, and nutrient losses [38]. Dialysis-induced nutritional stress tends also to reorientate liver protein synthesis toward acute phase proteins at the expense of albumin synthesis [39]. Furthermore, dialysis has been shown to induce hypercatabolism (increased energy expenditure) through the release of various stressors (hemodynamic and metabolic factors) [40] and mediators resulting from bioincompatibility reactions (protein cascade and cell activations) and blood–membrane interaction catalyzed by dialysis fluid impurity [41,42,43]. Factors that contribute to protein–energy wasting in dialysis patients are summarized in Figure 2.

Intercurrent-illness-related factors occur more frequently in dialysis patients as markers of vulnerability and contributing to their morbidity. They should be considered additional stressors and will not be discussed further. For example, they include various episodes of infection, cardiac event, vascular access problem, and surgical or any interventional acts.

## 3. Fluid Overload as a Cause of Protein Energy Malnutrition: Evidence-Based Facts

In this context, sodium and fluid imbalance, as well as their management, are emerging factors that contribute to protein–energy malnutrition. This is the main focus of this review to highlight this overlooked problem in dialysis patients. To facilitate description of the link between fluid disorders and protein energy malnutrition, we split this pathway into two parts, fitting with tides phenomena of intermittent hemodialysis. In brief, the tides up phenomenon reflects fluid and sodium accumulation occurring during the interdialytic period (as a result of diet and fluid intake and residual diuresis), while the tides down phenomenon reflects fluid and sodium depletion resulting from dialytic treatment (as a result of ultrafiltration and sodium depletion). Interestingly, both conditions are associated with the protein–energy-wasting process.

Chronic fluid overload (FO), a surrogate marker of water bound sodium excess, is quite common in hemodialysis patients. According to the method used to assess fluid status in dialysis patients (clinical, instrumental, biomarkers), FO prevalence may differ substantially. Using a non-invasive objective tool, such as multifrequency bioimpedance, to estimate fluid status, it was determined that FO is present in 40 to 60% of dialysis patients. In this context, it is interesting to note that FO prevalence is almost similar in hemodialysis and peritoneal dialysis patients, as shown in recent prospective international studies using the same MF-BIA tool [44,45,46,47]. Large retrospective cohort studies (MONDO Initiative) assessing FO with MF-BIA have consistently reported a higher mortality risk associated with FO in prevalent HD patients [48,49,50,51]. Furthermore, in these studies, it was also shown that the relative risk of death was positively associated with fluid overload degree independently from blood pressure levels and strongly linked to malnutrition inflammation complex syndrome [13,49,50,52]. These findings have been confirmed in a large international cohort study involving incident HD patients. As shown in this study, fluid overload at baseline is associated with 30% higher mortality risk at 12 months, increasing to 60% when FO was present one year later. Interestingly, mortality mainly from cardiovascular origins was associated with the degree of FO independently from blood pressure levels and worsened with higher time exposure [46]. These findings indicate that fluid overload is a common feature in dialysis patients, frequently associated with hypoalbuminemia, protein energy wasting process, and inflammatory markers, that highlights an unmet medical need [53].

Tissue sodium accumulation, as a surrogate marker of water free sodium stored in skin and muscle, is another common feature of kidney disease and dialysis patients recently identified as an additional vital risk factor [54]. Tissue sodium content, as measured via 23 Na MRI, is substantially increased in kidney disease and dialysis patients, with additional conditions that tend to worsen this condition (i.e., ageing, hypertension, cardiac, diabetes, salt diet, dialysate sodium) [55,56,57,58]. It is also shown in experimental and clinical studies that tissue sodium accumulation contributes via insulin resistance and inflammation to protein–energy wasting [56,59,60].

Hyponatremia, a well-known marker of poor outcome in hemodialysis patients [61] has been recently associated with severe fluid mixed disorders linked to intercurrent illnesses [48,49,62,63]. Additionally, it has been shown in a large Japanese study that patients’ outcomes (all-cause mortality, stroke, lower limb amputation) were worsened with positive plasma sodium changes and improved by negative plasma sodium changes [64]. These findings suggest that increasing fluid volume depletion in isotonic (isonatric) or hypotonic (hyponatric) dialysis conditions is preferable for improving outcome. As shown in a recent study monitoring natremia in hemodialysis patients using an automated sensor device, hypotonic hyponatremia was detected in about 10% of patients and in all cases was associated with a protein–energy-wasting and inflammatory process due to severe intercurrent illness [65]. Interestingly, correction of illness and readjustment of fluid status by significant reduction of dry weight up to 20% with isonatric dialysis were able in the majority of cases to mitigate risk, to correct hyponatremia, and to improve patients’ outcomes.

Clinical evidence linking fluid imbalance in dialysis patients with poor outcomes has been briefly summarized in this paragraph. Interestingly, all pathophysiologic pathways involved in these deleterious mechanisms are mediated by inflammation and associated with the protein–energy-wasting process [48].

## 4. Pathophysiologic Mechanisms Linking Fluid Overload, Fluid Management, and Protein Energy Malnutrition

While FO, inflammation, and malnutrition are independent risk factors for mortality [20,49,50,64], recent studies have shown that their combined presence may lead to a cumulative risk profile [46,66]. From a pathophysiologic perspective, FO, inflammation, and protein energy malnutrition can also be mutually reinforcing and can act both ways—from FO to inflammation and vice versa—suggesting that inflammation axis is the main cause of malnutrition [48]. Clinical evidence linking FO and inflammation has been recently summarized in a narrative review using objective tools to assess fluid status (multifrequency bioimpedance, cardiac biomarkers) and inflammation biomarkers (CRP, IL6) [53]. The pathophysiologic link is not clearly understood but likely involves several mechanisms: endothelial dysfunction and increased vascular cell adhesion molecules (VCAM-1); vascular leakage (increased capillary leak, angiopoietin 2); hypoalbuminemia resulting from reorientation of liver protein synthesis to acute phase proteins in case of inflammation [67,68]; muscle proteolysis due to reprioritization of cellular energy metabolism facing sodium accumulation or acute depletion [69,70]; gut endotoxin translocation (congestion or ischemia) [71]; pulmonary alveolar edema (breakdown of alveolar epithelial barrier) [72,73]; and finally tissue sodium storage activating immune and inflammatory pathways (water-free sodium, lymphocytes Th-17) [74,75,76] (see Figure 3).

Rapid sodium removal and fluid depletion as summarized by high ultrafiltration rate (>10 mL/h/kg) may lead to poor outcomes with increased cardiac risks [77,78]. This has been consistently shown in several studies suggesting that intensive or aggressive fluid management may lead to unwanted side effects. While these dialysis-induced risks are mainly mediated using hypovolemia and repetitive ischemic cardiac insults [69,79], some recent findings in animal experimental models suggest that fast sodium removal may trigger catabolism, with reprioritization of cell metabolism being associated with muscle proteolysis (release of free amino acids) to maintain tissue sodium content [70,80,81]. Fast sodium removal achieved through hemodialysis via ultrafiltration and negative dialysate–plasma sodium gradient is likely to contribute to muscle proteolysis and dialysis-induced protein catabolism [34,41,82]. In addition, as recently shown in a phosphate kinetic study relying on magnetic resonance spectroscopy, hemodialysis tends to preferentially deplete phosphate from the intracellular compartment, reducing the availability of high-energy phosphates and impairing ultimately mitochondrion and cellular metabolism [83].

Tissue sodium accumulation (skin, muscle) is associated with various pathophysiologic findings involving on one side the cardiovascular system (i.e., hypertension, left ventricular hypertrophy) independently from pressure level and mechanical consequences but on the other side various metabolic pathways that have direct effects on nutritional status (i.e., insulin resistance, muscle catabolism, cell energy metabolism) [54,56,84]. To reconcile the mismatch, the authors advocated surplus endogenous free water generation from exaggerated catabolic reactions and from enhanced renal accrual, which would make any extra exogenous water intake unnecessary [80,81].

Hyponatremia is difficult to explain in an anuric dialysis patient in which dialysis is the main source of exchange (sodium and water) with external milieu [61]. Several hypotheses have been advocated: Firstly, the release of mediators (vasopressin, angiotensin 2) [85] or factors (tonicity, thirst) that affect sodium-free water intake or retention; secondly, vascular leakage and sick-cell syndrome linked to inflammation that facilitate water intercompartmental imbalance [86,87]; thirdly, reorientation of liver protein synthesis to acute phase proteins, reducing albumin circulating levels. A new and interesting hypothesis may be formulated according to the findings in rodent models of tissue sodium content on muscle metabolism. As suggested by this model, muscle catabolism and renal recycling of urea in presence of tissue salt excess was found to be a key osmotic force in minimizing free water loss [70]. In the context of dialysis patients, tissue salt accumulation and depletion might be perceived as the main driving forces of tissue catabolism (proteins, carbohydrates, lipids) and oxidation mechanisms leading to increased endogenous production of metabolic water (sodium-free) [88]. In this case, hyponatremia will result from a salt-driven catabolic state, with muscle mass loss, enhanced proteins, carbohydrate and lipid breakdown, as well as from reprioritization of global energy muscle metabolism at the cellular level (mitochondrion) associated with an intense oxidation phosphorylation process leading to an excessive production and accumulation of endogenous metabolic water (sodium-free) [88].

In brief, these tides up and down phenomena, reflecting interdialytic fluid and sodium accumulation and intradialytic fluid and sodium depletion, respectively, are likely involved in the protein–energy malnutrition of hemodialysis patients [84]. During a tide up phenomenon, sodium and fluid accumulation tends to trigger inflammation and its related consequences (inflammation axis). During a tide down phenomenon, sodium and fluid depletion associated with stressors of dialysis-induced systemic stress tend to trigger catabolism and muscle proteolysis. As suggested in Figure 4, fluid management exposes hemodialysis patients to continuous protein–energy-wasting processes, a risk that can be mitigated by optimizing treatment schedule, increasing dietary intake, or promoting physical exercise.

## 5. Fluid Management and Correction of Fluid Imbalance

Optimal fluid and sodium management constitute a cornerstone component of dialysis adequacy in reducing cardiovascular burden in this fragile population. This is summarized in the dry weight probing approach [89,90]. Unfortunately, this basic component tends to be overlooked in favor of more attractive uremic toxins for improving cardiovascular burden. Dry weight policy in dialysis patients is a clinical priority that relies on the continuous and fine monitoring of patient conditions as well as anticipation when intercurrent event occurs (i.e., intervention, illness, infection) to preemptively readjust dry weight [91]. Fluid and sodium management represents a key component in dialysis patients working both ways—on one side to prevent fluid overload when secondary catabolism occurs and on the other side to prevent protein–energy wasting associated with chronic fluid overload [51]. Several recent reviews have been dedicated to this topic, to which we refer interested readers for more detailed information [54,92,93]. In this paragraph, we summarized the main points to address this problem in two sections: Firstly, how to monitor adequately fluid status of dialysis patients; secondly, how to restore fluid and sodium homeostasis. This is briefly schematized in Figure 4.

### 5.1. Monitoring Fluid and Sodium Status in Dialysis Patients

Clinical management is a global process that permits the attainment and maintenance of “dry weight” in dialysis patients [90]. However, it is a very subjective approach that takes advantage of using support tools for supporting clinical decision-making. In brief, dry weight must be reassessed and adjusted on a regular basis, taking into account the nutritional status of the patient as well as intercurrent events that may occur. In practice, frequent reassessment of dry weight (session-by-session to monthly) is recommended to follow patient clinical condition changes. Various non-invasive methods (offline or online HD machine) are currently proposed to assess fluid volume status and hemodynamic equilibrium. Chest X-ray may be used to assess lung overload and calculate cardiothoracic index (CTI). Inferior vena cava diameter (IVCD) assessed with a US probe is proposed to monitor intravascular volume and right atrial pressure at bedside in dialysis patients. However, its implementation is not easy, and its predictive value on hemodynamic stability is limited. Relative blood volume changes (RBVC), reflecting the vascular refilling rate (VRR), measured by an optical sensor are also proposed to estimate volume status. This tool may help to identify individual critical volume (risk of intradialytic hypotension) and to reduce the incidence of intradialytic hypotension. Long-term benefits still remain to be proved. Multifrequency bioimpedance (MF-BIS) is commonly used to assess fluid status and repartition in hemodialysis patients [94,95]. Several studies have confirmed that the MF-BIS is an easy-to-use, reliable, and reproducible tool for assessing volume status and relative fluid overload as well as body composition in dialysis patients. The lung ultrasound (LUS) method has also been proposed for assessing lung overload (extravascular edema) by measuring the thickness of interlobular septa. The thickening of interlobular septa via edema generates beams visualized as B lines (i.e., COMET tails) [73,96]. The simple numbering of these B lines provides an estimate of lung overload and cardiac dysfunction with highly predictive values of morbidity or mortality. Vascular stiffness estimated from pulse wave velocity (PWV) measurement provides an indirect way of assessing sodium content with highly predictive value on morbidity. Sodium MRI (23Na MRI) is proposed for assessing tissue sodium content (skin, muscles). This method remains a clinical research tool for evaluating the specific effects of new therapies or treatment regimens on tissue content [97,98]. Cardiac biomarkers are proposed for assessing myocardium structural or functional consequences associated with volume overload. Echocardiography is currently used to assess sodium overload and its cardiac consequences. Different echocardiographic criteria (volume of the atria, ventricular volume, left ventricular mass, thickness of the ventricular septum, ejection fraction, pulmonary arterial pressure) are recommended for the monitoring of dialysis patients. Atrial natriuretic peptides (ANP, BNP, and NT-proBNP) were the most widely used to assess volume overload [99]. Copeptin (a precursor of vasopressin) has recently been introduced to estimate volume depletion. Markers from the troponin family (troponins I and T) can be used to detect myocardial ischemic insults.

### 5.2. Restoration of Fluid and Sodium Homeostasis

Restoration of fluid and sodium imbalance relies on three main components: The first component is dietary counseling and educating patients to reduce sodium and fluid intakes with the support of specialized dieticians; the second component is based on the preservation of residual renal function. This aspect is fundamental but unfortunately will concern a limited number of patients over a short period of time; the third component relates to the treatment schedule program and dialysis prescription. For space reasons, only dialysis-related aspects will be discussed in this paragraph. Three items may be initiated to achieve this target: Firstly, weekly treatment time; secondly, ultrafiltration volume and rate; thirdly, dialysate sodium prescription.

Weekly dialysis treatment time is the main factor that conditions total ultrafiltered volume and ultrafiltration rate per session. Several studies have documented that an hourly ultrafiltration rate greater than 13 mL/kg was associated with an increased risk of mortality of almost 50% [78,100]. Preserving volemia should be a focused priority when establishing a dialysis treatment schedule [99]. Duration of dialysis sessions should be tailored to patients’ needs and tolerances to prevent reaching critical ultrafiltration rates and intradialytic hypotension [101]. This common sense approach is unfortunately not often applied for practical, logistical, or simply individual refusal reasons [100].

The prescription of dialysate sodium concentration is another important component in the prescription of dialysis that has been neglected [102,103]. The prescription of sodium dialysate responds in the majority of cases to a fixed prescription (sodium dialysate 138 or 140 mmol/l), to which the patient will adjust. Dialysate sodium prescription should be better individualized best based on a dialysate–plasma sodium gradient [82,93,104]. In fact, dialysate–plasma sodium gradient (d-pNa) conditions the diffusive sodium fluxes. Three clinical situations can be then observed: Firstly, the gradient is positive, in this case a diffusive sodium load is achieved resulting in sodium gain, reducing the net mass of sodium removed and increasing plasma tonicity during the dialysis session; secondly, the gradient is negative. In this case, a diffusive sodium removal is achieved, leading to sodium depletion while increasing the net mass of sodium removed and reducing plasma tonicity; thirdly, the gradient is neutral, in this case, there is no diffusive sodium transfer (isonatremic dialysis or zero diffusive condition), and the net sodium depletion relies only on convective transfers by ultrafiltration without plasma tonicity changes. It is important to highlight that the sodium mass removed in dialysis is mainly achieved via convection (80 to 100%) or ultrafiltration (intradialytic weight loss), while the diffusive part represents only a variable component ranging between 0 to 30% depending on the dialysate–plasma gradient. In brief, fine-tuning of a dialysate sodium prescription may be beneficial on sodium mass balance, tonicity changes, and dialysis tolerance. Automated dialysate sodium alignment is facilitated by means of a sodium management module embedded on a certain dialysis machine that permits an automated adjustment of dialysate sodium to plasma sodium concentration according to the prescription. In addition, this sodium management module gives sodium mass removed and plasma sodium concentration in real time [105,106,107].

As a brief summary, it is time to take fluid balance and nutrition seriously, particularly in dialysis patients, with appropriate measures to reduce protein–energy wasting burden in this fragile population [108].

## 6. Conclusions

As highlighted in this in-depth narrative review, sodium and water imbalance and fluid management in dialysis patients tend to be strongly associated with the protein–energy-wasting process. Fluid and sodium imbalance in dialysis patients are superimposed factors that tend to precipitate highly malnutritional exposed patients. Inflammation, and its related pathways, is the main source of protein energy wasting. Tissue sodium accumulation and depletion, as well as their metabolic consequences, tend to worsen nutritional conditions. Hyponatremia, a condition associated with poor outcomes, is strongly associated with mixed water and sodium disorders. We hypothesize that hyponatremia reflects intense hypercatabolism, resulting in metabolic water production from cellular oxidation phosphorylation processes. It is time to act more precisely on sodium and fluid imbalance to mitigate both cardiovascular and nutritional risks and to improve patient-reported outcomes. Personalized management of sodium and fluid using available tools, including sodium management tools, has the potential to more adequately restore sodium and water homeostasis and to improve the nutritional status of dialysis patients. In brief, fluid and sodium imbalance must be considered to be strong determinants of the protein–energy-wasting process requiring more precise approaches. This new hypothesis deserves further interventional clinical and nutritional studies to be validated.

## Figures and Tables

**Figure 1 nutrients-14-04489-f001:**
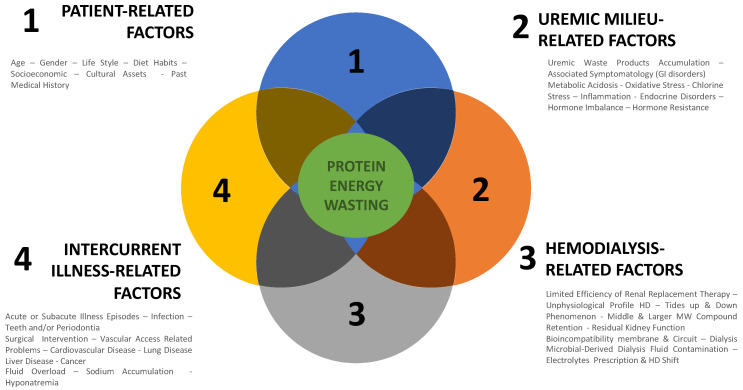
Causes of protein–energy malnutrition in dialysis patients.

**Figure 2 nutrients-14-04489-f002:**
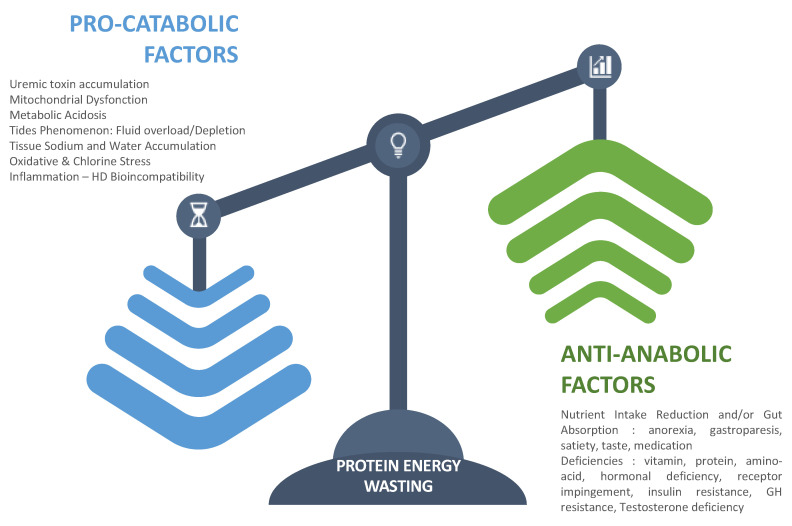
Clustering factors that favor the protein–energy-wasting process.

**Figure 3 nutrients-14-04489-f003:**
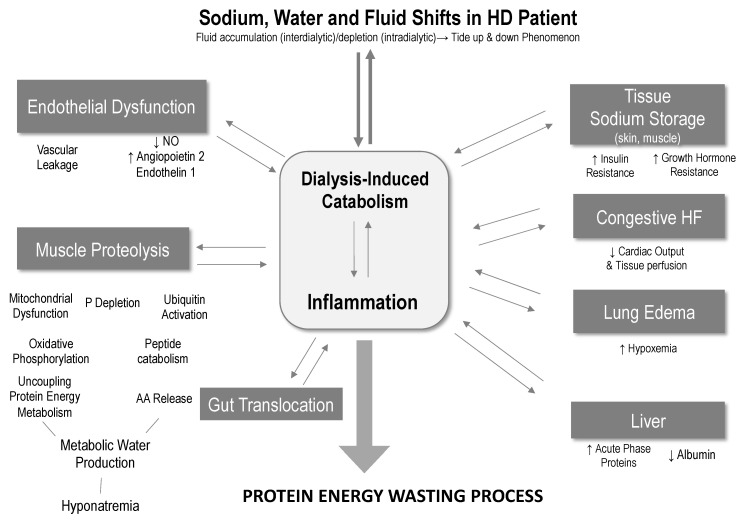
Proposed pathophysiologic link between sodium, fluid volume, and PEM.

**Figure 4 nutrients-14-04489-f004:**
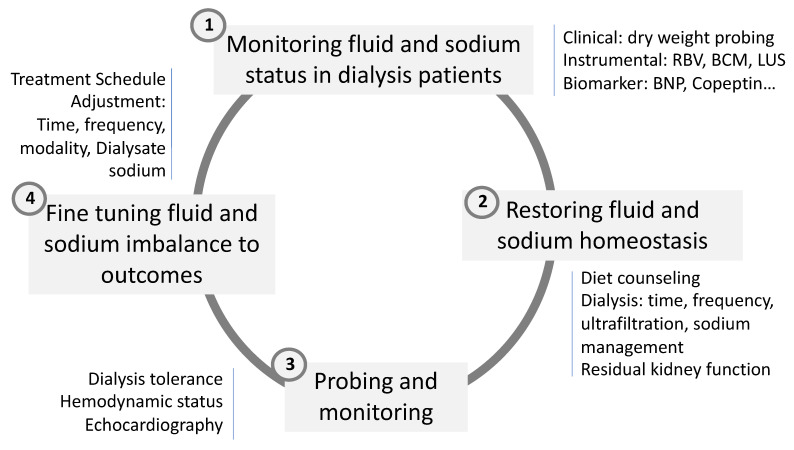
Four clinical steps involved in sodium, water, and fluid volume management in HD patient.

## Data Availability

Not applicable.

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
