# Peer review of "Fluid Overload and Tissue Sodium Accumulation as Main Drivers of Protein Energy Malnutrition in Dialysis Patients"

_nutrients, 2022, doi:10.3390/nu14214489_

Round 1
Reviewer 1 Report
The manuscript is the review on PEW and fluid (sodium/water) overload (FO), which I found is very extensive and good for the busy readers to grasp the current concept and understanding on the topic. However, I would rather like to give some suggestions for possible revisions.
* The title focus on FO affecting PEW in dialysis. However, the chapter 1 (introduction) and 2 are extensive review on PEW irrespective of FO, which needs to be more concise and focused more on facts related to FO.
*Chapter 3 is titled "FO as a cause of PEW; evidence based facts", however, the discussion and referenced evidence are on FO as a cause of patient prognosis (mortality and CV events). Authors should offer more evidence on FO as a cause of "PEW" but not general prognosis.
*In the discussion on tissue Na accumulation (chapter 3), no mention about PEW is found, which needs to be more discussed and evidence on it should be offered.
*In the discussion on reference #60 (chapter 3), authors stated that "increasing sodium mass depletion ///// is preferable for improving outcomes", however, I cannot understand what this means or why it can be said so.
* In chapter 4 regarding tissue Na accumulation and mechanism linking with PEW is hard to understand. As authors refer to reference #78, in chronic kidney disease (CKD) model, aestivation-like phenomena occurs because CKD lose renal Na loss, which activated water preservation response, involving muscle catabolism to produce urea. How this or its related mechanism occur in patients with no functional kidney.
* Most of the statements in chapter 5 is related with general management of fluid (Na/water) management in dialysis, which have been extensively discussed in other literatures. I would like this chapter to mainly discussed on fluid management, which especially merits on prevention of PEW.
Author Response
Thanks for your positive feedback and useful comments. A precise point by point answer is attached.

Author Response
Thanks for your very positive feedback encouraging comments
Reviewer 3 Report
This is a nicely written and timely review that links two subjects that occupy nephrologists since the beginning of the dialysis era: protein energy wasting and salt and water retention/removal. Although the hypothesis that these two entities are truly linked still needs more data in humans before general acceptance, the article sums up several arguments to support their hypothesis.
In the same line, they formulate some interesting insight on the pathophysiology of hyponatremia in dialysis patients. The article is written in a fluid style, and easy to read. Basic research concepts are mixed with clinical observations.
Therefore, I have no major remarks. However, I would appreciate if the authors would mention that the fluid overload-malnutrition hypothesis, albeit appealing, still needs interventional studies to show that better fluid manageemtn improves nutritional status and halts catabolism. As for the sodium management module in some modern dialysis machines, here too, randomized studies will be needed to demonstrate definite benefit. These studies should also integrate QoL parameters.
But all together, nice piece of work, it was a pleasure to read it.
Author Response
Thanks for your positive feedback and useful comments. A more detailed answer is attached.

Round 2
Reviewer 1 Report
Authors responded to the comments satisfactorily. No further comments.